# Switchable Terahertz Absorber from Single Broadband to Dual Broadband Based on Graphene and Vanadium Dioxide

**DOI:** 10.3390/nano12132172

**Published:** 2022-06-24

**Authors:** Guan Wang, Tong Wu, Yang Jia, Yang Gao, Yachen Gao

**Affiliations:** Electronic Engineering College, Heilongjiang University, Harbin 150080, China; wang2687220886@163.com (G.W.); 124wutong@163.com (T.W.); yang_1990@163.com (Y.J.); gaoy_hit@163.com (Y.G.)

**Keywords:** terahertz, graphene, vanadium dioxide, impedance matching, absorption

## Abstract

A multifunctional switchable terahertz (THz) absorber based on graphene and vanadium dioxide (VO_2_) is presented. The properties of the absorber are studied theoretically by the finite-difference time-domain (FDTD) method. The results illustrate that the structure switches between the single-broadband or double-broadband absorption depending on the temperature of VO_2_. Moreover, the amplitude of the absorptivity can be adjusted by changing the Fermi energy level (*E_F_*) of graphene or the conductivity of VO_2_ separately. Via impedance matching theory, the physical mechanism of the absorber is researched. Furthermore, the effects of incidence angle on absorption have also been studied. It is found that the absorber is insensitive to the polarization of electromagnetic waves.

## 1. Introduction

A THz wave is an electromagnetic wave with a frequency ranging from 0.1 THz to 10 THz. It has received much attention in the application prospects of communication [1,2], biology [3], sensing [4], imaging [5,6], and other fields. The promotion of THz technology is limited because of the lack of suitable materials in nature. Metamaterials, as special electromagnetic response materials, are composed of sub-wavelength microstructures produced by artificial designs [7,8]. Using the combination of metamaterials and THz technology, researchers have designed a variety of high-efficiency optical devices such as filters [9,10], modulators [11], switchers [12], and absorbers [13,14]. Among them, THz absorbers based on metamaterials have shown promising applications in the fields of invisibility cloaks, wireless communications, and thermal emitters [15,16,17]. The absorption properties of traditional THz metamaterial absorbers (TMMAs) cannot be easy to change once it is produced, which leads to inconvenience and high cost. Therefore, metamaterials that can be dynamically controlled have become a hot research topic [18,19].

Graphene is a new material, a two-dimensional honeycomb lattice structure with tightly packed carbon atoms connected by sp^2^ hybridization. It has become a hotspot in both physics and chemistry for its unique electro-optical properties [18,20]. The *E_F_* of graphene can be tuned continuously and dynamically by varying gate voltage or chemical doping, which makes graphene be widely studied in tunable TMMAs [21]. For instance, in 2020, Liu et al. presented a THz absorber which realized broadband tunable function by changing the *E_F_* of the graphene [20]. In 2021, Feng et al. showed a tunable wide-band THz absorber that can be tuned by varying *E_F_* of graphene. The absorptance can be gradually reduced by decreasing the *E_F_* [22]. In recent years, phase change materials have also been used in a wide range of tunable devices. VO_2_ is a phase change material. When the temperature reached the critical temperature of 340 K, it changes from insulating phase to metal phase, and the phase change can be reversible [23,24]. Hence, VO_2_ has great application potential for developing devices with tunability functions. Researchers have studied different tunable absorbers based on VO_2_. For instance, in 2020, Huang et al. designed a flexible wideband THz absorber composing of VO_2_ square rings. The absorptance can be tuned in the range between 4% and 100% by varying the conductivity of VO_2_ in the range from 200 S/m to 2 × 10^5^ S/m [19]. In 2021, Wu et al. proposed an ultra-wideband absorber consisting of three VO_2_ resonance rings. By switching between the insulating and metallic phases of VO_2_, the absorber can realize the function of an optical switcher, and the working band reaches the range of 2.34 to 5.64 THz [25].

For realizing more functions, the multifunctional modulation absorbers achieved via the combination of graphene and VO_2_ are also the focal direction. In 2020, based on a multilayer structure of graphene and VO_2_, Zhang et al. presented a THz multifunctional absorber that can achieve both narrowband and broadband absorption properties [26]. In 2021, Liu et al. proposed a structure based on graphene and VO_2_ to realize switchable single broadband and double narrowband absorption [27]. In 2021, Liu et al. realized THz absorbers with broadband characteristics at different frequencies based on hybrid metamaterials of graphene and VO_2_ [28]. We can see that the studies above focused mainly on narrowband absorption or single broadband properties.

In the paper, we proposed a THz absorber that can realize switching from single-broadband to double-broadband absorption by using graphene and VO_2_ as controlling media. Via the FDTD solution method, we theoretically studied the absorption properties and mechanisms of the designed metamaterials. 

## 2. Structure and Method

The designed multifunctional tunable absorber is shown in Figure 1a, in which every structure unit in the device is arranged periodically. The three-dimensional schematic diagram of the structure unit is shown in Figure 1b. It can be found from Figure 1b that the unit cell consists of six layers. From top to bottom, they are the VO_2_ metamaterial patch, polyethylene cyclic olefin copolymer (topas), graphene patch layer, VO_2_ film, topas, and gold (Au) film, respectively. The thicknesses of the upper and lower topas layers are t_1_ = 18 μm and t_2_ = 16 μm, respectively, and the relative permittivity of topas is ε = 2.35 [29]. The thickness of the underlying Au film is 0.5 μm, and its conductivity is 4.561 × 10^7^ S/m [30]. The thickness of the top VO_2_ metamaterial patch and the middle VO_2_ layer is 0.1 μm and 2 μm, respectively. The top VO_2_ metamaterial patch is shown in Figure 1c. It is composed of two orthogonal hexagons with structural parameters w_1_ = 4 μm, w_2_ = 10 μm, and *l*_1_ = 32 μm. The top view of the graphene patch layer is presented in Figure 1d. The side length of the square graphene patch is *l*_2_ = 15 μm, and the distance between the two square patches is w_3_ = 2 μm. The graphene patch is linked with a graphene bar w_4_ = 1 μm wide to form a conductive pathway when a voltage is applied.

Then, we begin to calculate reflection parameter *S*_11_ and transmission parameter *S*_21_ via the FDTD method. The x and y directions are set to be periodic boundary conditions and the z-direction is set to be a perfectly matched layer. The absorptance of the structure can be expressed as: A(ω)=1−R(ω)−T(ω). T(ω) represents the transmittance of the absorber, R(ω) represents the reflectance of the absorber, and $R(\omega )={\left|S_{11}(\omega )\right|}^{2}$, $T(\omega )={\left|S_{21}(\omega )\right|}^{2}$. As the thickness of Au film on the bottom of the structure is much larger than the skin depth of THz wave, T(ω)=0. Therefore, the absorption rate is $A(\omega )=1-R(\omega )=1-{\left|S_{11}(\omega )\right|}^{2}$.

The surface conductivity of graphene can be described as $\sigma_{g}=\sigma_{\mathrm{int}ra}+\sigma_{\mathrm{int}er}$. Where $\sigma_{\mathrm{int}ra}$ and $\sigma_{\mathrm{int}er}$ represent the intraband and interband contributions, respectively. According to random-phase approximation (RPA) theory $\sigma_{g}$ can be expressed specifically as [31,32]:(1)$$\sigma_{\mathrm{int}ra}=\frac{2e^{2}k_{B}T}{\pi \hbar^{2}}\frac{i}{\omega +i\tau^{-1}}\ln[2\cosh(\frac{E_{F}}{2k_{B}T})]$$
(2)$$\sigma_{\mathrm{int}er}=\frac{e^{2}}{4\hbar }[\frac{1}{2}+\frac{1}{\pi }\arctan(\frac{\hbar \omega -2E_{F}}{2k_{B}T})-\frac{i}{2\pi }\ln\frac{{\left(\hbar \omega +2E_{F}\right)}^{2}}{{\left(\hbar \omega -2E_{F}\right)}^{2}+4{\left(k_{B}T\right)}^{2}}]$$
where e is the electron charge, $k_{B}$ is the Boltzmann constant, T is the temperature, ħ is the reduced Planck’s constant, ω is the frequency of THz wave, τ is the relaxation time of graphene carrier, and $E_{F}$ is the Fermi level.

Based on the Drude model, in the THz band the insulating constant of VO_2_ can be defined as [33,34,35,36]:(3)$$\varepsilon (\omega )=\varepsilon_{\infty }-\frac{\omega_{p}^{2}(\sigma )}{\omega^{2}+i\gamma \omega }$$
where $\varepsilon_{\infty }=12$ is insulating permittivity at the infinite frequency, $\omega_{P}(\sigma )=1.4\times 10^{15}$ rad/s is plasma frequency depending on conductivity, and $\gamma =5.75\times 10^{13}$ rad/s is collision frequency. The relationship of $\omega_{P}(\sigma_{0})$ and conductivity σ is: $\omega_{p}^{2}=\omega_{p}^{2}(\sigma_{0})\sigma /\sigma_{0}$, with $\sigma_{0}=3\times 10^{5}$ S/m. The conductivity of VO_2_ is 0 S/m or 2 × 10^5^ S/m when it is insulating or metal phase.

## 3. Results and Discussion

Based on the formula mentioned above, via FDTD the absorptance spectra of the absorber were obtained and shown in Figure 2, where the blue solid line and red dashed line represent the absorptance when VO_2_ is in the insulating and metallic phases, respectively. As shown with the blue solid line in Figure 2, when VO_2_ is in the insulating phase, the absorber shows a broadband absorption ranging from 0.9 THz to 3.5 THz, and with the absorptance of more than 90%, which results from the single-cycle characteristic of the graphene layer. When the temperature of VO_2_ reaches phase-change temperature (68 °C), VO_2_ behaves as a metal and has large conductivity (about 2 × 10^5^ S/m) [37], which makes the structure show double-cycle characteristics. Correspondingly, the absorber shows two broadband absorptions, and the absorptance is more than 90% in the regions of 1.5–3.6 THz and 7.1–8.5 THz.

For explaining the physical mechanisms of the designed absorber, we investigated the electric field distributions when VO_2_ is in the insulating phase and the $E_{F}$ of graphene is set to be 0.7 eV. The electric field distributions of the graphene layer are shown in Figure 3a,b, respectively. It can be found that, at 1.2 THz and 3 THz, the electric field is concentrated mainly in the gaps between the square graphene sheets, which implies the resonance of the graphene layer mainly occurs in the interaction between the squares of graphene.

Further, we investigated the electric field distribution in the top layer when the temperature is high and the VO_2_ is in the metallic phase. The distribution of electric fields at 1.8 THz and 3.4 THz are depicted in Figure 4a,b. We can see that the electric field focused mainly on the left and right ends of the VO_2_ patch, which means that the resonance of two frequencies results from the electro-dipole excitation of the left and right ends of the VO_2_ patch. The electric field distribution at 7.3 THz and 8.3 THz are depicted in Figure 4c,d. The enhanced electric field is at the sidewalls of the hexagonal patch and two ends of the patch, which implies the resonance of two frequencies arise here.

In addition, the absorption properties of single broadband and dual broadband were studied respectively. Specifically, the effects of the $E_{F}$ of graphene and the conductivity of VO_2_ on the absorption spectra are shown in Figure 5a,b, respectively. When VO_2_ is in the insulating phase, it can be found in Figure 5a that the absorptance decreases significantly with the decrease of the $E_{F}$. In Figure 5b, it seems that as the conductivity of VO_2_ decreases, the absorptance of the dual broadband becomes gradually low.

In order to further analyze the results above, the impedance matching theory is utilized, and the corresponding impedance matching formula is as follows [38,39]:(4)$$A(\omega )=1-R(\omega )=1-{\left|\frac{Z-Z_{0}}{Z+Z_{0}}\right|}^{2}=1-{\left|\frac{Z_{r}-1}{Z_{r}+1}\right|}^{2}$$
(5)$$Z_{r}=\pm \sqrt{\frac{{\left(1+S_{11}(\omega )\right)}^{2}-S_{21}{\left(\omega \right)}^{2}}{{\left(1-S_{11}(\omega )\right)}^{2}-S_{21}{\left(\omega \right)}^{2}}}$$
where A(ω) is the absorptance, R(ω) is the reflectance, Z is the effective impedance, $Z_{0}$ is the free space impedance, and $Z_{r}=Z/Z_{0}$ is the relative impedance between the designed structure and the free space. When $Z_{r}=1$ the absorptance is maximal. The $S_{11}(\omega )$ and $S_{21}(\omega )$ are the reflection coefficient and transmission coefficient obtained from S-parameters. When the bottom Au film is much thicker than the skin depth of the THz wave, the transmittance T(ω) will be close to 0. The real and imaginary parts of the relative impedances $Z_{r}$ are calculated by utilizing the MATLAB programming package. Under the different $E_{F}$ of graphene, the real and imaginary parts of the relative impedances $Z_{r}$ are represented in Figure 6a,b. In Figure 6a, there are three peaks and four dips, and they approach gradually to 1 with increasing $E_{F}$. When the $E_{F}$ reaches 0.7 eV, the real part is almost equal to 1 in the region of 0.9–3.5 THz. In Figure 6b, the three peaks and three dips are gradually close to 0 with the increase of $E_{F}$. At the $E_{F}$ = 0.7 eV, the imaginary part is almost equal to 0 in the region of 0.9–3.5 THz. This means that the proposed absorber matches well with the impedance of free space. Correspondingly, it can be seen from Equation (4) that the absorptance reaches the maximum. Similarly, the change of real and imaginary parts of the relative impedances $Z_{r}$ with the conductivity of VO_2_ is described in Figure 6c,d. In Figure 6c, the peaks and dips of the curve become close to 1 with the increase in conductivity. When the conductivity increases to 2 × 10^5^ S/m, the curve is almost equal to 1 at 1.5–3.6 THz and 7.1–8.5 THz. Figure 6d shows that all the peaks and dips move gradually near to 0 as the conductivity increases, and the imaginary part is closest to 0 at 1.5–3.6 THz and 7.1–8.5 THz for 2 × 10^5^ S/m. When the conductivity is 2 × 10^5^ S/m, the impedances of the absorber and free space match perfectly at 1.5–3.6 THz and 7.3–8.5 THz. From Equation (4), we can know that when the impedance matches perfectly the absorptance reaches the maximum. 

In addition, we also studied the effects of incidence and polarization angles on absorptance. The incident wave is parallel to the xoz plane. Here, the incident angle was between the incident direction and the negative direction of the *z*-axis. The absorption performance spectra of different polarization angles for single and dual broadband are shown in Figure 7a,d, respectively. The absorption performance is insensitive to polarization angle, which may result from the fact that the proposed structure is a four-sided symmetrical structure. Figure 7b,c show the single-broadband absorptance performance spectra of TE and TM waves changing with incidence angle. It can be found that the absorptance remained almost constant even when the incidence angle is 50°. Figure 7e,f are the absorption spectra of TE and TM waves of dual broadband. It can be found that the absorptance is not sensitive to the incident angle of TE and TM polarized plane waves. In the lower frequency region, the absorptance is still 80% when the incident angle is 50°. When the angle of incidence is greater than 50° the absorptance decreases gradually. With an increase in the incident angle, a significant blue shift occurs in the high frequency of dual broadband, but the absorptance is still high. Based on the analysis above, we can know that the absorber can realize good-broadband absorption performance even at large incidence angles.

At present, the absorbers utilizing VO_2_ and graphene as modulating media have attracted increasing research [26,27,28], and every investigation has its novelty. To more visually compare their advantages, Table 1 shows the characteristics of the proposed absorber and other absorbers. The absorber in Ref. [26] can realize tunable narrowband and broadband absorption properties. In Ref. [27], the absorber can implement the switching from dual narrowband to single-broadband absorption. In Ref. [28], the absorber can realize the switching between high-frequency broadband and low-frequency broadband. However, our work can allow for flexible switching between single and double broadband via VO_2_. In addition, the amplitude of absorptance can be tuned by using graphene and VO_2_.

## 4. Conclusions

In summary, we propose a dynamically tunable THz absorber that can switch from single-broadband absorption to double-broadband absorption by using VO_2_ and graphene. When VO_2_ is in the insulating phase and the $E_{F}$ of graphene is 0.7 eV, the absorber exhibits single-period characteristics, resulting in single-broadband absorption. When VO_2_ is in the metallic phase, the absorber exhibits dual-period characteristics, thus achieving dual-broadband absorption. The amplitude of the absorbance can also be tuned individually. Additionally, the absorbance is insensitive to incident wave polarization.

## Figures and Tables

**Figure 1 nanomaterials-12-02172-f001:**
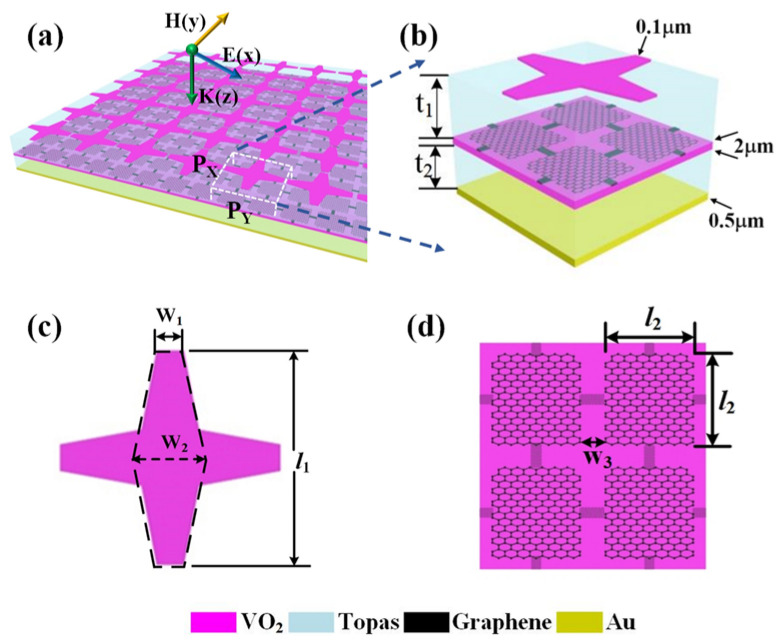
(**a**) The overall diagram of the designed structure; (**b**) Three-dimensional image for the unit cell; (**c**) The top view of the top VO_2_ metamaterial patch; (**d**) The schematic diagram of the graphene patch layer.

**Figure 2 nanomaterials-12-02172-f002:**
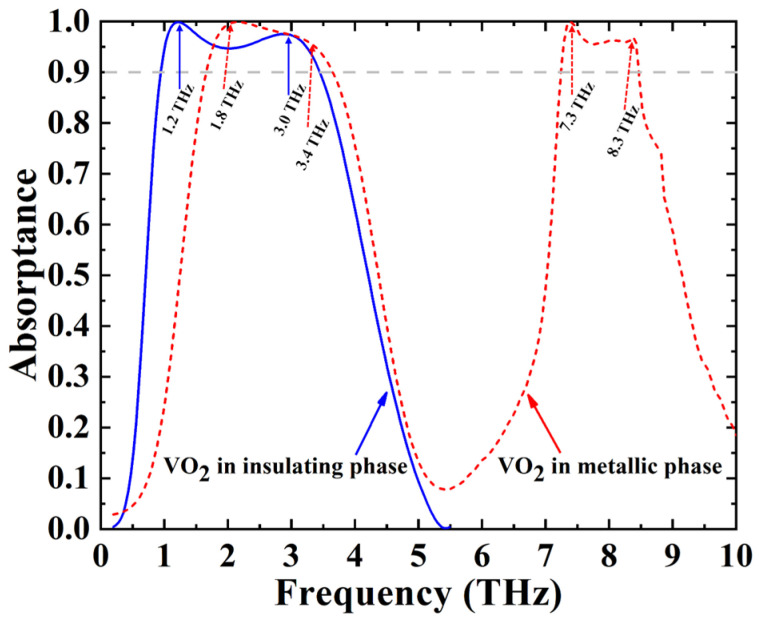
Absorptance spectrum of the absorber in two different modes.

**Figure 3 nanomaterials-12-02172-f003:**
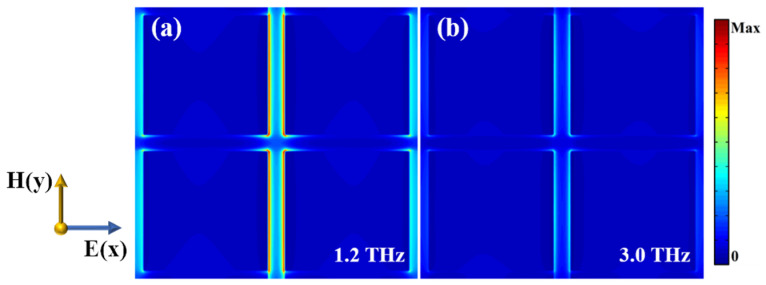
Electric field distribution of graphene patch layer in xoy plane at (**a**) 1.2 THz, and (**b**) 3 THz.

**Figure 4 nanomaterials-12-02172-f004:**
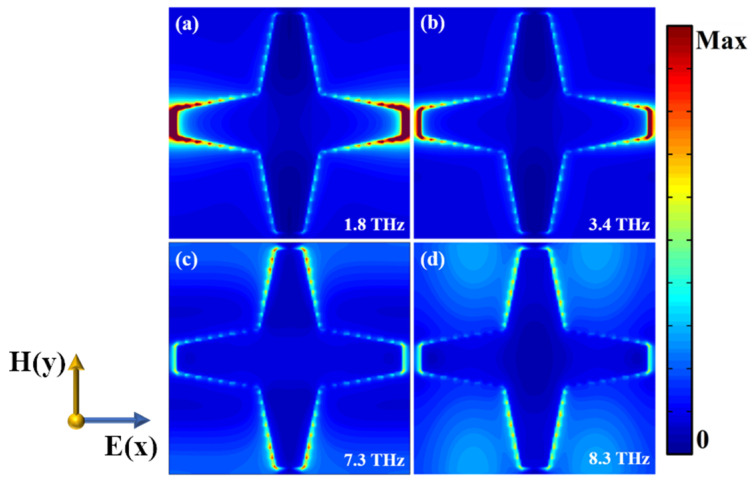
Electric field distribution of the top VO_2_ patch in the xoy plane at (**a**) 1.8 THz, (**b**) 3.4 THz, (**c**) 7.3 THz, and (**d**) 8.3 THz.

**Figure 5 nanomaterials-12-02172-f005:**
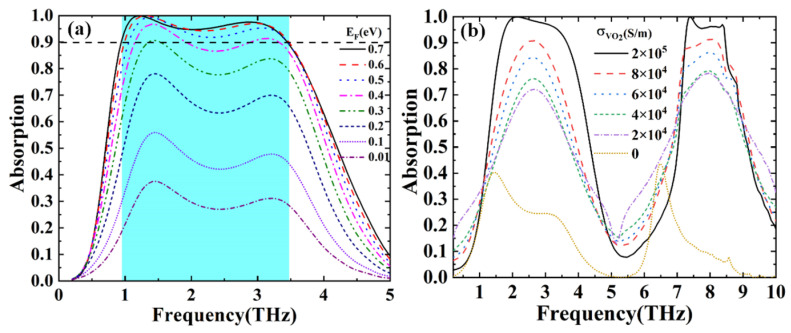
(**a**) Broadband absorptance spectra with different $E_{F}$; (**b**) Dual broadband absorptance spectra with different conductivities.

**Figure 6 nanomaterials-12-02172-f006:**
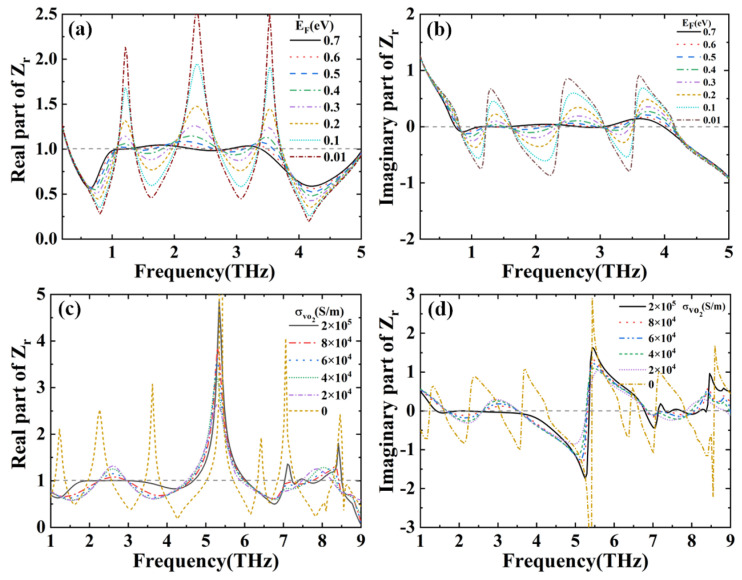
(**a**) Real and (**b**) imaginary parts of relative impedance $Z_{r}$ for different $E_{F}$; (**c**) Real and (**d**) imaginary parts of relative impedance $Z_{r}$ for different VO_2_ conductivities.

**Figure 7 nanomaterials-12-02172-f007:**
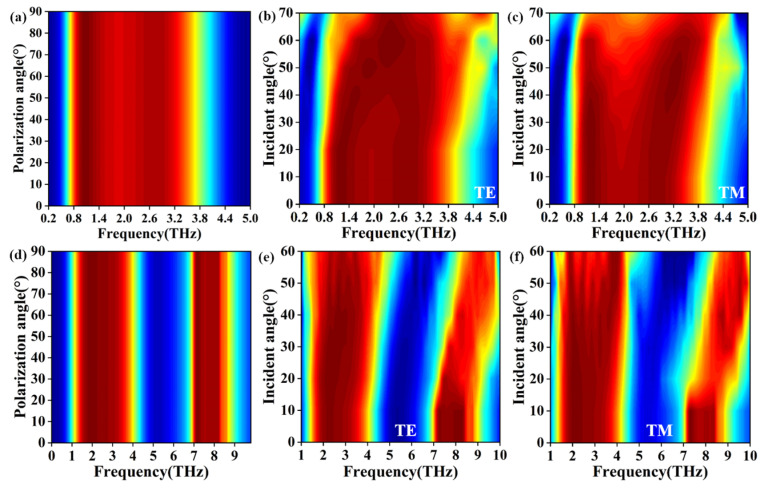
(**a**) Single-broadband absorption spectrum with changing plane wave polarization angle; (**b**) TE and (**c**) TM polarization with single-broadband absorption at different incidence angles; (**d**) Double-broadband absorption spectrum with changing plane wave polarization angle; (**e**) TE and (**f**) TM polarization with dual-broadband absorption at different incidence angles.

**Table 1 nanomaterials-12-02172-t001:** The absorber in this paper is compared with other tunable absorbers in the THz band based on VO_2_ and graphene.

Reference	Adjustable Material	Absorption Band	Function
[26]	VO_2_ & Graphene	Narrow & Broad	Narrowband and broadband switching & Tuned absorptance
[27]	VO_2_ & Graphene	Dual narrow & Broad	Dual narrowband and single-broadband switching & Tuned absorptance
[28]	VO_2_ & Graphene	High frequency broad & Low frequency broad	High-frequency broadband and low-frequency broadband switching & Tuned absorptance
This work	VO_2_ & Graphene	Single broad & Dual broad	Single-broadband and dual-broadband switching & Tuned absorptance

## Data Availability

All content and data have been displayed in the manuscript.

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
