# Peer review of "Switchable Terahertz Absorber from Single Broadband to Dual Broadband Based on Graphene and Vanadium Dioxide"

_nanomaterials, 2022, doi:10.3390/nano12132172_

Round 1

Reviewer 1 Report

After a careful reviewing of your manuscript, I need to inform that It is not unable to be publish this manuscript in present form in the Nanoscale. The paper id absolutely out of scope this journal. 

Author Response

Thank you very much for your review! In our proposed structure, the thickness of vanadium dioxide patch on the top layer is 100 nm and the thickness of the underlying gold film is 500 nm, so the size of the structure is in the nanoscale. We think it is in the scope of “Nanomaterials”, especially in the Nanocomposite section.

The designed absorber can switch between the single-broadband and double-broadband absorption depending on the conductivity of vanadium dioxide. when vanadium dioxide is in insulating phase, the absorber shows a single-broadband absorption ranging from 0.9 THz to 3.5 THz, and with the absorptance of more than 90%. When vanadium dioxide is in metallic phase, the absorber shows two broadband absorptions, and the absorptance is more than 90% in the regions of 1.5-3.6 THz and 7.1-8.5 THz. Moreover, the absorptance can be adjusted by changing the Fermi energy level of graphene or the conductivity of vanadium dioxide, separately. Furthermore, the absorption performance is insensitive to angle of polarization and incidence. The absorptance remains almost constant even when the angle of incidence is 50°, and the absorptance still remains 80%. The physical mechanism of the absorber was analyzed by using impedance matching theory. The structure provides a new view for the development of terahertz devices in the nanoscale. We hope fervently that you can reconsider it.

Reviewer 2 Report

The manuscript “Switchable terahertz absorber from single broadband to dual broadband based on graphene and vanadium dioxide” by Guan Wang, Tong Wu, Yang Jia, Yang Gao and Yachen Gao presents a design of a metamaterial absorber characterized by broadband absorption in the terahertz frequency range. The specific phase properties of vanadium dioxide (VO2) and graphene, resulting from the possibility of adjusting the energy of the Fermi level, according to the authors, allow the construction of an effective switchable absorber. Such an absorber below 10 THz is characterized by a single absorption band for the temperature range corresponding to the VO 2 non-conducting phase and a double absorption band above the phase transition temperature to the conducting phase. The authors analyzed the influence of the Fermi level in the graphene layer and the conductivity of the VO2 layer on the absorption value in the considered bands. The presented results may arouse interest due to the potential applications of this type of devices. A separate issue that the authors do not address is the possibility of producing structures according to the proposed design.

Basically, the manuscript has the correct layout and is well-edited. Some shortcomings relate to the following issues:

  1. The descriptions of the dimensions in Figure 1 are too small and therefore cluttered.
  2. The graphene layer shown in Figure 1 has a marked structure, which may suggest that this was assumed in the model.
  3. The authors did not specify which programming package they used in the analysis of the properties of their absorber.
  4. The formula on line 94 of the manuscript was unnecessarily repeated on line 97.
  5. Figures 3 and 4 lack information about the direction of the electric field vector of the probe wave.
  6. There is no data on the authors of publication 33 in the references.

After appropriate corrections have been made, I can recommend the manuscript for publication in Nanomaterials (MDPI).

Round 2

Reviewer 1 Report

Authors presented good research on VO2 THz absorber. Reviewer didn’t find the correspondence to the scope of hte Nanometerials. (Reminer:Nanomaterials are materials with typical size features in the lower nanometer size range and characteristic mesoscopic properties; for example quantum size effects. These properties make them attractive objects of fundamental research and potential new applications. The scope of Nanomaterials covers the preparation, characterization and application of all nanomaterials.

The following examples may provide a guide to what will be covered (not exclusive):

  • Nanomaterials:
    • Nanoparticles, coatings and thin films, inorganic-organic hybrids and composites (i.e. MOFs), membranes, nano-alloys, quantum dots, self-assemblies, graphene, nanotubes, etc
  • Methodologies:
    • Synthesis of organic, inorganic, and hybrid nanomaterials
    • Characterization of mesoscopic properties
    • Modelling of nanomaterials and/or mesoscopic effects
  • Applications:
    • Any application of new nanomaterials or new application of nanomaterials)
    •  

This work does not offer significant new nanomaterials or mesoscopic properties insights that would be interest to the broad of nanomaterials readership especially with respect to putting the work in proper context through appropriate consideration of other published work. Manuscript should be presented to more specified readers.